# Good Agricultural Practices Related to Water and Soil as a Means of Adaptation of Mediterranean Olive Growing to Extreme Climate-Water Conditions

Nektarios N. Kourgialas [1,*], Georgios Psarras [1], Giasemi Morianou [1], Vassilios Pisinaras [2], Georgios Koubouris [1], Nektaria Digalaki [1], Stella Malliaraki [3], Katerina Aggelaki [3], Georgios Motakis [4] and George Arampatzis [2]

[1] Water Recourses-Irrigation & Environmental Geoinformatics Laboratory, Institute of Olive, Tree Subtropical Crops and Viticulture, Hellenic Agricultural Organization (ELGO Dimitra), 73134 Chania, Greece

[2] Soil and Water Resources Institute, Hellenic Agricultural Organization (ELGO Dimitra), 57001 Thessaloniki, Greece

[3] Agricultural Cooperative Partnership Mirabello Union S.A., 72400 Neapoli, Greece

[4] K.E.DH.P. Platanias Municipality Development Enterprise, 73134 Chania, Greece

\* Correspondence: kourgialas@elgo.gr

**Abstract:** Despite the fact that the olive tree is one of the best-adapted species in Mediterranean hydroclimate conditions, climate extremes impose negative effects on olive fruit set and development and subsequently on crop yield. Considering that the frequency of climate extremes is increasing in the last years due to climate change, Good Agricultural Practices (GAPs) have to be applied in order to mitigate their impact on olive trees. In this context, 18 experimental olive groves (irrigated and rainfed) were established, located on the island of Crete (south Greece). A set of 13 GAPs were applied in different combinations, mainly targeting to reduce water losses and erosion, alleviate heat stress and increase water use efficiency. Each experimental orchard was divided into two parts, the control (business-as-usual) and experimental (GAPs implementation). Four indicators were used for the assessment of GAPs performance, namely, Water Productivity (WP), Economic Water Productivity (EWP), Runoff (RF), and Yield (Y). WP and EWP were found to be up to 2.02 and 2.20 times higher, respectively, in the demonstration part of the orchards compared to the control, while Y was found to be up to 119% higher. RF was higher up to 190% in the control compared to the demonstration part of the experimental orchards. The above results clearly demonstrate that the implementation of the proposed GAPs can significantly support the adaptation of olive crops to extreme conditions.

**Keywords:** olive trees; agricultural practices; water efficiency; crop yield

## 1. Introduction

The olive tree (*Olea europaea* L.) is one of the most vital fruit trees in the Mediterranean region [1,2]. The cultivation of olive trees is an ancient tradition, with great importance in the economy and the environment [3,4]. Greece is the third largest producer of olive oil in the world, after Spain and Italy [5]. The annual olive oil production at a country level ranges between 300,000–400,000 tons during high cropping years, while the olive oil produced on the island of Crete represents about 30% of this amount [6]. Olive trees could thrive on different types of soil, at different altitudes (from sea level up to 1200 m a.s.l.), and under various climate conditions [4]. Although the olive tree is considered one of the best-adapted species and can tolerate high levels of water stress, its shoot growth, and fruiting could be affected under extreme climate-water conditions [7]. Extreme weather conditions attributed to climate change, such as heatwaves and intensive rainfalls, have been recorded frequently during the last years, imposing negative effects on crop yield of both rainfed and irrigated olive trees in the Mediterranean basin [8].

In order to mitigate climate-water effects on tree fruiting, Good Agricultural Practices (GAPs) related to water and soil need to be adopted by the farmers. So far, limited studies



assess the impact of the implementation of a set of proper agricultural practices on olive cultivation [5]. Bechara [9] pointed out that sustainable management practices with low environmental impact, such as minimum tillage or no-tillage, application of recycled organic matter, pruning, proper irrigation, and fertilization are important in order to improve soil physicochemical characteristics, water efficiency, crop productivity, reduce soil erosion by water and environmental pollution. Specifically, the practices of no-tillage, weed mowing, and cover crops have positive effects on the soil microbial community (increasing its microbial biomass), promote flora biodiversity, reduce $CO_2$ emissions triggered by soil tillage, increase soil water storage, and provide protection against soil erosion [5,10–14]. Michalopoulos [5] also mentioned that proper pruning techniques, that enhance the within-canopy light distribution, the aeration of the foliage, and the development of bearing shoots, could contribute to the reduction of the "alternate bearing" phenomenon (high/low yield year) and the achievement of stable yearly crop yield. Furthermore, the same study proposes that shredding of pruning residues, instead of burning, increases soil organic matter, providing both a mulching layer and improved soil water retention capacity. In addition, the negative effects of excessive heat periods during summer may be reduced by spraying and covering leaves with a thin layer of different compounds [15–17]. Kaolin sprays applied to drought-stressed olive trees can reduce canopy temperature, heat stress, and sunburn impacts [15,16]. Moreover, kaolin offers a repelling effect against major olive pests like olive fruit flies [18,19].

Orchard management and especially irrigation management in arid and semi-arid regions require special attention in order to increase water use efficiency, optimize crop productivity and reduce environmental impacts (e.g., degradation of ground and surface water, and soil erosion). Water Productivity (WP) is an important indicator of irrigation water use, described with many different equations and definitions in the literature [20–24]. The latest scientific publications that have tried to give a clear and complete definition of this indicator propose that Water Productivity (WP) in agricultural irrigation may be defined as the ratio between the actual crop yield achieved and the total water involved in crop production [25,26]. However, Fernández [27], suggested that the user should consider that the WP approach alone is not enough for safe conclusions. Their findings show that decision-making on cultivation and irrigation management improves significantly when physical and economic water productivity indicators are combined with information derived from both the production and profit functions of the crop. The economic value of water is of great importance but so is the economic return that results for the farmer, therefore the Economic Water Productivity (EWP) consists also an important indicator [21,25,26]. Economic Water Productivity is defined as the value of the actual Yield (€) divided by the total water used [26].

In the present study, a set of Good Agricultural Practices (GAPs) was implemented, including irrigation according to crop needs, no-tillage, cover crops, winter & summer pruning, weed control only by mowing during spring and summer, mulching of pruning residues, application of spray compounds (kaolin), proper fertilization (based on leaf and soil analyses) and application of compost. Natural barriers, strategically located vertical to the water flow, were also introduced to reduce water runoff and erosion at selected farms in sloppy areas. The efficiency of the GAPs at the field scale by considering and quantifying the WP and EWP indicators was also examined. Crop yield was quantified and presented, as an indicator that expresses the desired result in the context of the sustainability of olive cultivation under extreme climatic conditions.

The aforementioned water-use indicators could be adopted for any orchard management, irrigated or non-irrigated, and ensure more efficient use of the water and conservation of water resources under scarcity conditions. In order to have a complete picture of water savings in the field, especially for sloping olive parcels which occupy a substantial portion of the Mediterranean semi-mountainous landscape, we should also take into account factors that are considered as the non-consumed fraction, such as the runoff and the drainage [27]. Thus, one of the aims of this study was the development of a field Runoff

(RF) performance indicator for measuring water runoff at sloping olive parcels to quantify the possible positive effects of the proposed GAPs. Specifically, GAPs including the natural barriers were evaluated in terms of runoff reduction and soil moisture retention based on water traps that were constructed on the soil of sloping parcels to measure the amount of water runoff. Considering that in many agricultural Mediterranean areas extreme climate conditions call into question the sustainability of olive cultivation, the objective of this work was to evaluate and compare, for the first time, using key water-use indicators, a set of GAPs as a holistic framework of adaptation approaches to extremely dry or wet conditions for Mediterranean olive growing.

## 2. Materials and Methods

### 2.1. Study Area

In the eastern part of the Mediterranean basin, Crete is the largest island in Greece with a total area of about 8265 km$^2$. The eastern part of Crete includes the prefectures of Heraklion and Lassithi, while the western includes the Chania and Rethymno prefectures. The altitudes in Crete range from 0 m to 2400 m above sea level. 26% of the total island area, is covered by lowlands (<200 m), 56% is covered by semi-mountainous areas (200–800 m) and about 18% is covered by mountainous land (>800 m) (Figure 1). The climate in Crete is the sub-humid Mediterranean with humid and quite cold winters, and warm and dry summers. The annual rainfall ranges from 300 to 700 mm/year in the plains, from 700 to 1000 mm/year in the semi-mountainous areas, while in the mountainous areas it reaches 2000 mm/year. According to the CORINE Land Cover maps (2018), the agricultural areas in Crete cover about 3205 km$^2$. These areas are primarily dominated by tree crops (olive, citrus, avocado), greenhouse vegetable cultivations, and vineyards. Tree crops cover about 72% of the total agricultural area in Crete [28]. Olive trees and vineyards thrive in the eastern part of the island due to the local drier climate conditions. On the other hand, the agricultural areas in the western part of the island are dominated by olive, citrus, and avocado cultivations [29,30]. Olive growing is the dominant cultivation in Crete, occupying significant areas of the lowland (0–200 m) and almost all the cultivation areas of the semi-mountainous region (200–800 m) (Figure 1).

Due to the complex topography of the island, important climatological differences occur between the eastern and western parts, with the eastern part of the island receiving lower amounts of precipitation compared to the western [31]. The main water demand in Crete is for irrigation purposes (81% of the total water consumption), while the main source of water comes from groundwater (92% of the total water supply). Since the agricultural sector is the primary consumer of water in Crete, applying appropriate agricultural practices for protecting the water-soil resources and ensuring productivity at the same time, is a great task [32]. Considering that extreme hydrological events (drought/flood) appear with increasing incidence in Crete, detailed knowledge of the potential positive effects of proper agricultural practices to save water and decrease the impact of future water-climate effects are essential in ensuring agricultural production in prone climate-water agro-environments. In this study, we applied GAPs for improving water-soil conservation in olive growing areas (the dominant cultivation in Crete), in two pilot river basins, one on the western side and one on the eastern side of the island during the period of three years (2017–2019), as part of LIFE AgroClimaWater project. For the two studied river basins, 18 experimental olive groves were selected to apply the GAPs (Figure 1). These experimental sites were selected based on different criteria (climatic, geomorphological/landscape, soil characteristics, water availability, and the already applied agricultural practices) to capture an extensive range of agricultural ecosystems in Crete, including soil, climatic, and cropping systems heterogeneity. Table 1 describes the main statistical values of the soil properties (soil texture, organic matter, bulk density, and Ph) of the 18 experimental parcels. Based on this table, 13 out of the 18 experimental parcels are classified in the medium texture class, 2 are classified in the fine soil texture class, and the remaining 3 experimental parcels are defined in the coarse texture class. The above proportion of the experimental fields per soil texture

class corresponds to the percentages of olive orchards present in the various soil types of Crete [31]. Also, mean and SD (standard deviation) values of organic matter, bulk density, and Ph are noted in Table 1. All these values are in agreement with the corresponding typical values of soil types where the Cretan olive groves are located.

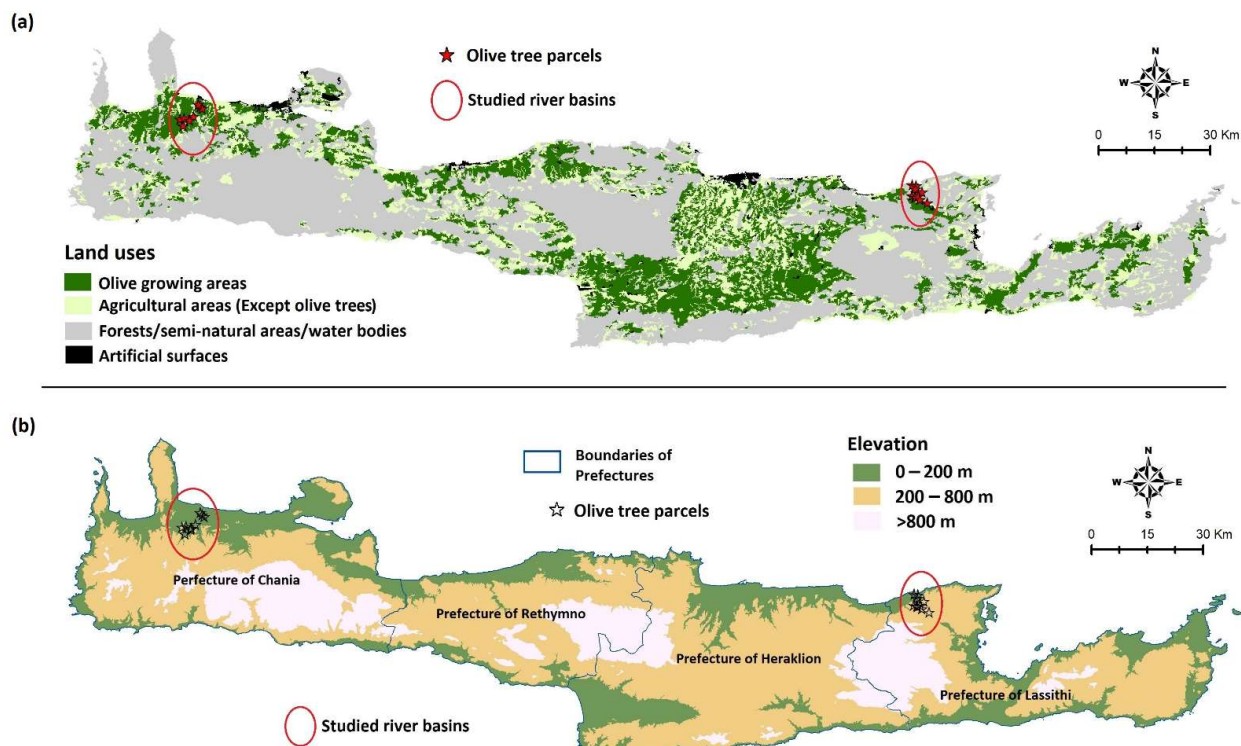

**Figure 1.** Maps depicting (**a**) Land uses in Crete, (**b**) the geomorphology and the prefectures of Crete. In both cases the experimental olive orchards (parcels) in the two pilot areas are represented.

**Table 1.** Soil properties of the 18 experimental parcels.

| Soil Texture Class | Number of Parcels | Mean Organic Matter (%) | Mean Bulk Density (g/cm³) | Mean Ph |
|---|---|---|---|---|
| Fine | 2 | 4.36 (SD: 0.28) | 1.33 (SD: 0.003) | 7.70 (SD: 0.57) |
| Medium | 13 | 3.86 (SD: 1.29) | 1.41 (SD: 0.026) | 7.55 (SD: 0.38) |
| Coarse | 3 | 4.36 (SD: 0.84) | 1.47 (SD: 0.040) | 7.70 (SD: 0.44) |

*2.2. Experimental Design and Performance Indicators*

GAPs have been implemented during 2017, 2018, and 2019 (3 implementation years) in all 18 experimental olive groves (cultivation density: 200 trees per ha). Figure 2 depicts the experimental design for each studied farm (non-sloping and sloping farms). Each pilot farm was divided into two sections: (a) the control part and (b) the demonstration part, [Control (T1), Demonstration (T2)] where different treatments were applied. In the Control treatment (T1) common/traditional management by the farmer was applied, while in the Demonstration treatment (T2), covering a fixed area of 0.2 ha, GAPs were applied. Our aim was, through comprehensive in situ monitoring approaches, during the three implementation years, to examine the effectiveness of the proposed GAPs by comparing the results between Control (T1) and Demonstration (T2) treatment for the establishment of olive pilot farms adapted to extreme climate conditions. For evaluation purposes, two sub-groups of Irrigated (Irr.) and Rainfed (NO-Irr.) farms were established.

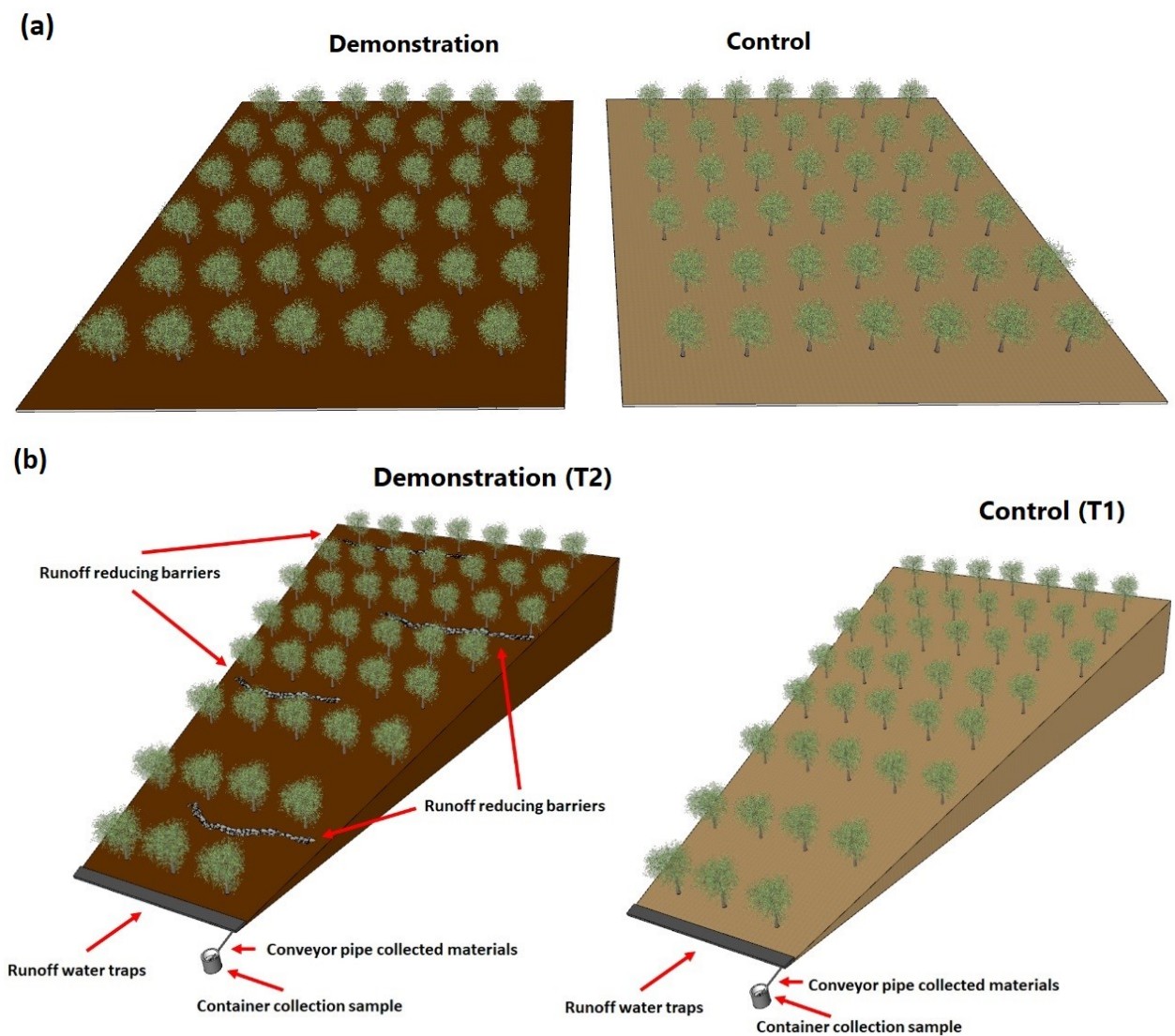

**Figure 2.** Experimental design for (**a**) non-sloping parcels, and (**b**) sloping parcels.

Plant protection was common in both parts of the field, following the typical practices applied by the farmer, not differentiated between treatments. Seven (7) out of eighteen (18) studied farms were irrigated and the remaining eleven (11) were rainfed. This proportion corresponds to the ratio of 6:4 of rainfed vs. irrigated olive orchards in Crete, while within each pilot area the ratio was different representing the higher ratio of irrigated olive orchards in Western Crete as compared to the Eastern part of the island. Since in Crete drip irrigation is the main irrigation system used for olive cultivation [32], the irrigation system used in both treatments of this study was drip irrigation. For the Demonstration (T2) treatment the irrigation needs were determined by the FAO56 version of the Penman equation using the meteorological data recorded by weather stations near the orchards. Moreover, soil moisture data were used to validate the need for the applied irrigation dose proposed by the Penman equation. Furthermore, in cases of dry periods and low water availability, irrigation was applied only at critical stages of plant growth (deficit irrigation). Based on the above, the amount of water supplied through irrigation for each parcel was variable depending on climatic conditions, soil parameters, water availability, crop phenological stages, and soil moisture dynamics of each parcel. Also, according to the olive cultivation geomorphology in Crete, four out of the total 18 studied orchards were located in sloping areas (slope above 6%) (Figure 2b).

The GAPs implemented at Demonstration (T2) parcels as well as their positive effects on water saving are summarized in Table 2.

**Table 2.** The GAPs implemented in the Demonstration (T2) treatment plots with remarks on the expected positive effects.

| Practice | Remarks |
|---|---|
| No weed mowing during winter | Reduce water losses from Runoff/Increase water infiltration into the soil/Reduce water erosion |
| No soil tillage | Reduce water erosion |
| Weed mowing during spring and summer | Reduce water losses by transpiration of weeds/reduce water losses by evaporation due to soil mulching |
| Introduction of cover crops–grass and legume mix | Reduce runoff water losses/Increase water infiltration into the soil/Reduce erosion |
| Winter pruning | Reduce water losses by transpiration |
| Summer pruning | Reduce water losses by transpiration |
| Shredding of pruning | Reduce water losses by evaporation (mulching) |
| Application of organic material (compost) | Increase water holding capacity/Reduce water losses through deep percolation |
| Application of aluminum silicate (kaolin–Surround® WP, Phoenix, AZ, USA) | Reduce heat stress/improve leaf functioning and water use efficiency |
| Application of fertilizers based on leaf and soil analyses (winter application) | Reduce mineral nutrient leaching/reduce groundwater pollution |
| Application of fertigation & foliar application of fertilizers | Increase the efficiency of fertilizing/reduce mineral nutrient leaching |
| Drip irrigation according to meteorological data and soil moisture data | Rational use of irrigation water/increase water use efficiency |
| Construction of runoff-reducing barriers | Reduce runoff water losses/Increase water infiltration into the soil/ Reduce water erosion |

Based on the above-applied GAPs, the aim was to optimize all the specific components of the hydrological cycle at the farm scale as described in Figure 3, so that inputs are increased, and losses are reduced. This approach in turn can lead to an increase in water use efficiency resulting in higher yield using less water.

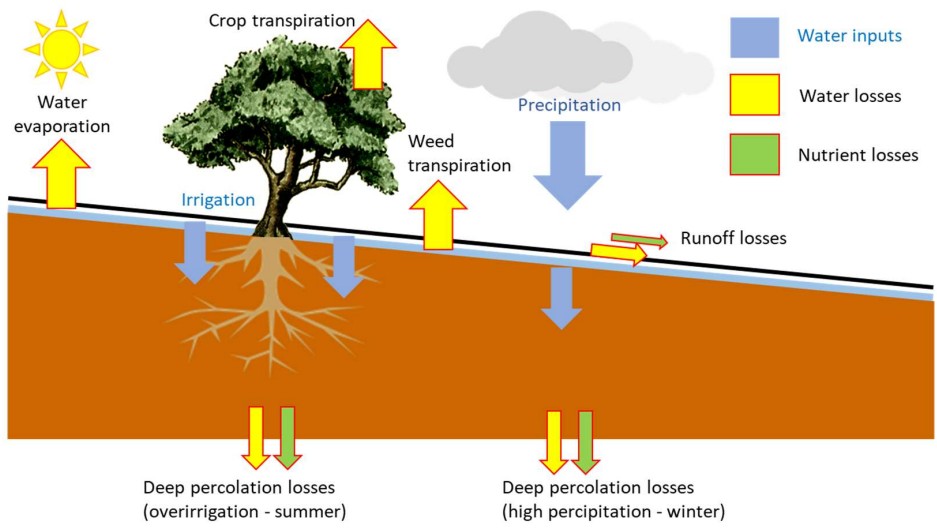

**Figure 3.** The components of the hydrological cycle at farm scale.

For the Control (T1) and the Demonstration (T2) treatments as well as for Irrigation (Irr.) and NO Irrigation (NO Irr.) strategy, in order to evaluate the results of the suggested GAPs for the three-year implementation (2017, 2018 and 2019) the following three water use performance indicators as well as the actual crop yield indicator were estimated:

### 2.2.1. Water Productivity (WP)

Water Productivity [kg m$^{-3}$] is commonly defined as the ratio between the crop yield achieved (Y) expressed in kg ha$^{-1}$ over the total water use for the irrigation period from May to October (TWU = irrigation + rainfall) expressed in m$^3$ ha$^{-1}$ [26]:

$$WP = Y/TWU \tag{1}$$

WP indicator expresses the benefit derived from water consumption and can be utilized for evaluating the impact of on-farm practices, especially under scarce water conditions. It provides an indication of the water that could be saved with the best-achieved crop yield [33].

### 2.2.2. Economic Water Productivity (EWP)

Economic Water Productivity [€/m$^3$] is defined as the ratio between the economic value of the marketable olive yield (olive yield [kg ha$^{-1}$] × % olive oil × % marketable product × price of product [€ kg$^{-1}$]) expressed in kg ha$^{-1}$ over the total water use for the irrigation period from May to October (TWU = irrigation + rainfall) expressed in m$^3$ ha$^{-1}$ [26]:

$$EWP = (\text{olive yield} \times \% \text{ olive oil} \times \% \text{ marketable product} \times \text{price of product})/TWU \tag{2}$$

EWP indicator has the benefit that incorporates, apart from water use and yield, essential parameters such as product value (farmer's income) and olive oil content.

### 2.2.3. Runoff (RF)

As depicted in Figure 4, both schematically and in field photos, at sloping parcels, water traps were introduced in order to evaluate the efficiency of the runoff barriers to increase water infiltration into the soil. Specifically, strategically located water traps were established at an outlet (endpoint) location of the drainage area of 0.2 ha for each Control (T1) and Demonstration (T2) treatment, capturing the surface runoff. The amount of water collected through the traps was regularly recorded after each major rainfall event, resulting in the calculation of the RF indicator expressed in m$^3$ of water per treatment area and per rainfall event.

### 2.2.4. Yield

The Actual Crop Yield Achieved was also Determined in this Study as a Simple but of High Importance Indicator. Actual Olive Yield (Kg ha$^{-1}$) was Compared between Control (T1) and Demonstration (T2) treatments for Irrigation (Irr.) and NO Irrigation (NO Irr.) Strategy.

It is very important to mention that for Crete, 2017 was an average year in terms of climatic conditions, 2018 was an extremely dry year, and 2019 was an extremely wet year. Specifically, the average rainfall during 2017, 2018, and 2019 for the parcels studied in Crete was recorded at 583.5 mm, 376.5 mm and 815.0 mm respectively. Also, the average air temperature during 2017, 2018, and 2019 for the parcels studied were recorded at 19.2 °C, 20.6 °C, and 18.8 °C, respectively (data from meteorological stations in the studied areas). Thus, a challenge of this work was to investigate, through the above-mentioned indicators, whether the implementation of the proposed GAPs could have positive effects on the sustainability of Cretan olive groves under different or even extreme climatic conditions. For this reason, data analysis to obtain the three water use performance indicators as well

as the actual crop yield indicator was applied separately for each of the three years to show the effectiveness of the GAPs under different climatic conditions.

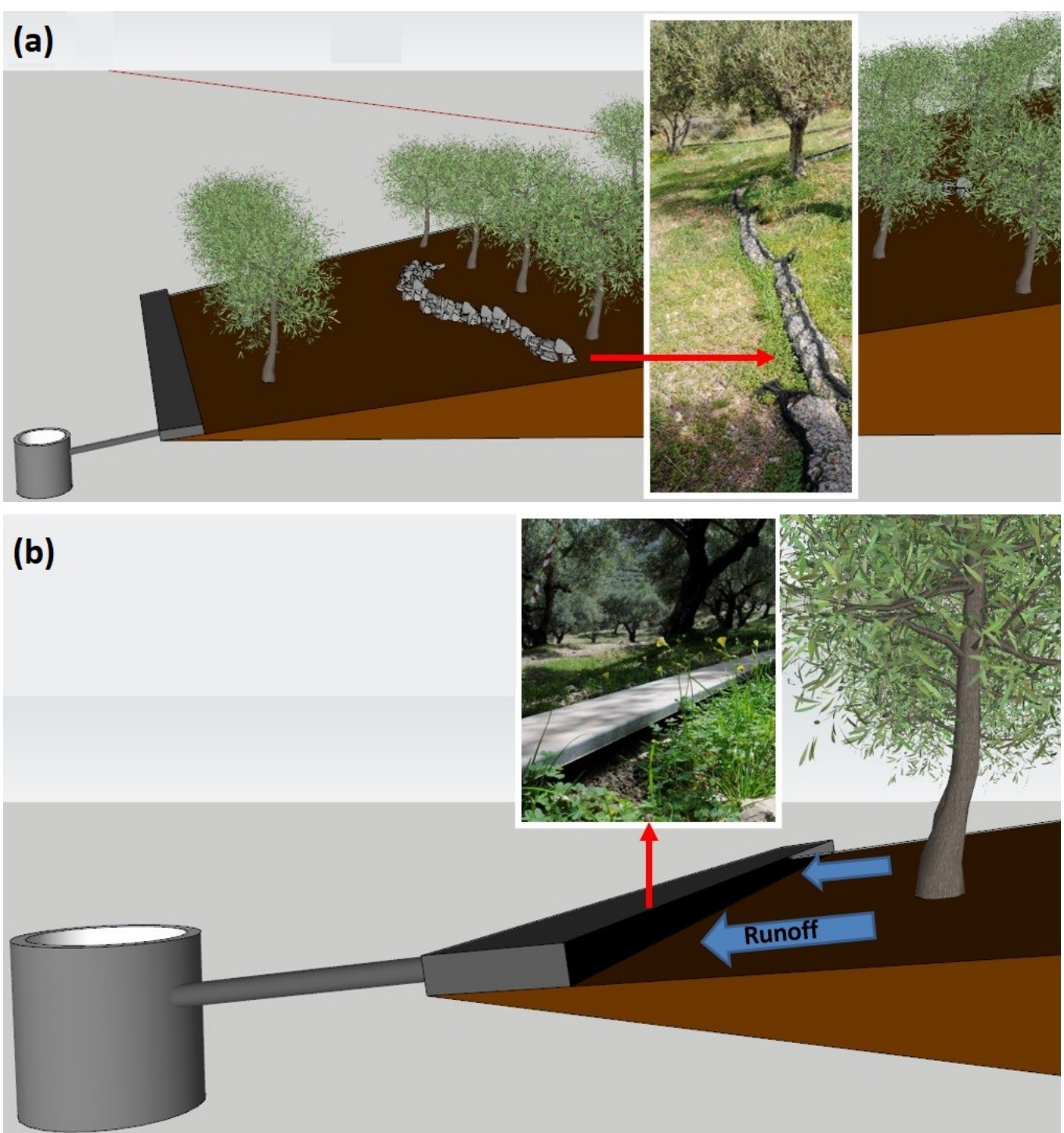

**Figure 4.** (**a**) Runoff reduction of natural barriers in the field, (**b**) Surface runoff collection system (traps) in the field.

### 2.2.5. Statistical Analysis

Statistical analysis was performed using the SPSSv17 package (SPSS Inc., Chicago, IL, USA) with one-way analysis of variance (ANOVA), of two treatments (T1, T2) split for Irrigation (Irr.) and NO Irrigation (NO Irr.) strategy. Eighteen replicate parcels per treatment were used concerning the WP, EWP, and Y indicators, while four replicate parcels

per treatment were used for the RF indicator. When there was a significant difference (*p*-value < 0.05), means were separated using Duncan's test.

## 3. Results and Discussion

In general, Crete could be considered one of the main "Hot Spots for Climate Change" in the Mediterranean area, meaning that this area is especially susceptible to climate change. Based on the proposed GAPs, a crucial and challenging task of this work was to increase the adaptability of olive trees to extreme climate conditions. Challenges that had to be met during the implementation of this work were the extremely dry and relatively hot year of 2018, as well as the extremely wet and milder in terms of summer temperatures year of 2019 that led to intense crop losses by pests and diseases. Including the typical year of 2017, the experimental work of this case study captured the average as well as different extreme climate conditions that are expected to increase in intensity over time. Proving the efficiency of the proposed GAPs scheme under these quite variable conditions would be a quite promising result for farmers to adapt to the challenges introduced by climate change [5].

*Yield and Water Use Performance Indicators*

Table 3 summarizes the results of the studied Yield and Water use performance indicators. Values are the mean of eighteen replicate parcels per treatment concerning the WP, EWP, and Y indicators, and four replicate parcels per treatment for the RF indicator. Values followed by different letters within a column are significantly different (sd) and values followed by the same letter are not significantly different (nsd) at the 0.05 level of probability, according to Duncan's test.

**Table 3.** Water and yield-related indicators during the experiment period for the control (T1) and the Demonstration (T2) treatments for Irrigation (Irr.) and NO Irrigation (NO Irr.) strategy. The indicators analyzed are Water Productivity (WP), Economical Water Productivity (EWP), Runoff (RF), and Yield (Y).

| Indicators | Treatment | (2017) Irr. | (2017) NO Irr. | (2018) Irr. | (2018) NO Irr. | (2019) Irr. | (2019) NO Irr. |
|---|---|---|---|---|---|---|---|
| WP | T1 | 5.85 c | 2.50 ab | 2.02 a | 2.26 a | 4.68 bc | 6.71 cd |
|  | T2 | 6.39 c | 2.69 ab | 3.02 ab | 4.56 bc | 5.89 c | 8.53 d |
|  |  | nsd | nsd | nsd | sd | nsd | nsd |
| EWP | T1 | 4.40 cd | 2.13 a | 2.03 a | 2.94 ab | 2.23 ab | 4.36 cd |
|  | T2 | 3.39 bc | 2.47 ab | 4.48 cd | 5.24 de | 4.48 cd | 5.93 e |
|  |  | nsd | nsd | sd | sd | sd | sd |
| RF | T1 | 6.0 cd | 5.8 c | 1.6 ab | 2.9 ab | 9.3 ef | 11.7 f |
|  | T2 | 3.7 bc | 6.4 cd | 0.7 a | 1.0 ab | 6.3 cd | 8.7 de |
|  |  | nsd | nsd | nsd | nsd | sd | sd |
| Y | T1 | 9694.4 d | 4406.3 ab | 8156.1 cd | 2628.6 a | 2445.5 a | 2348.4 a |
|  | T2 | 8303.0 cd | 3585.0 ab | 9969.4 d | 5748.8 bc | 3236.0 ab | 3123.8 ab |
|  |  | nsd | nsd | nsd | sd | nsd | nsd |

Values followed by different letters within a column are significantly different (sd) and values followed by the same letter are not significantly different (nsd) at the 0.05 level of probability, according to Duncan's test.

The results of Yield shown in Table 3 are within the range presented in a recent Mediterranean scale review analysis for various density olive groves in response to water supply [27]. The yield was higher in the demonstration part of the orchard (T2) as compared to the control part (T1) during the years 2018 and 2019, although statistically different only in the rainfed orchards during 2018 (Table 3). Non-statistically significant differences in yield are reasonable due to the fact that there is a significant variation in the potential yield of different orchards due to differences in tree age, soil fertility, and alternate bearing cycles among the different orchards. Therefore, despite the non-statistical differences, the fact

that yield was higher in T2 during years 2 and 3 by 22 to 32% in irrigated orchards and by 33 to 119% in rainfed orchards indicates a positive trend (Figure 5). This is supported by the fact that 2018 and 2019 were two years with high and low extremes in precipitation, although it should not be underestimated that 2017 was the first year of GAPs application, and trees needed time to respond to the alterations in orchard management. Data from the three years indicate the impact of extreme events on the yield of olive trees in control parcels. In T1, the yield was highest during 2017, but it was reduced during the dry and wet years of 2018 and 2019, respectively. On the contrary, in T2, the highest yields were recorded during the driest year (2018) which is a promising fact given that water scarcity is one of the most significant threats even for oliviculture under the predicted climate change scenarios for Crete [34]. Both T1 and T2 treatments gave the lowest yields during 2019. This reflects the great impact of combined pest and disease problems that significantly reduced olive oil production in Crete during that year [35]. Although pest and disease control was not included in the GAPs schedule applied in the present study, yield in T2 was less affected than in T1. This can be due to the indirect effect of the product that was applied for reducing heat stress (kaolin) since it is also used for reducing olive fruit fly damage [36].

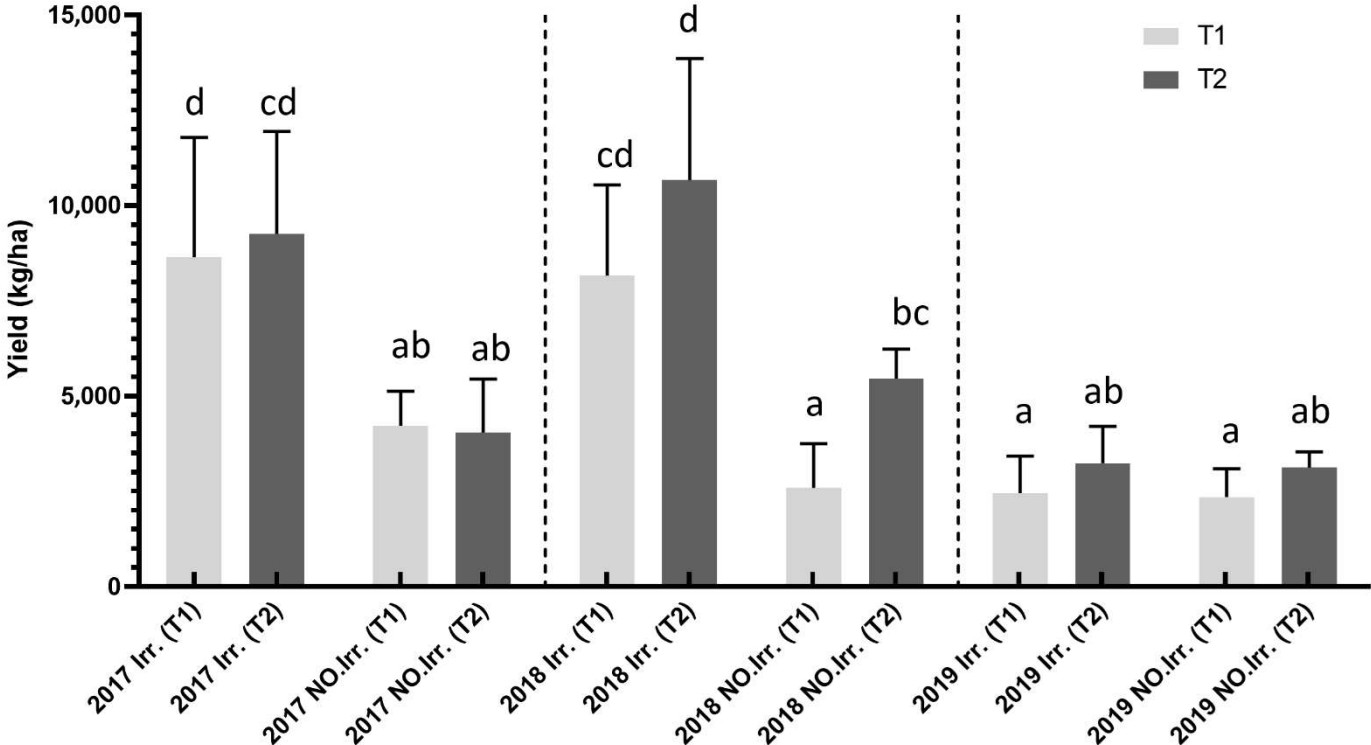

**Figure 5.** Yield (Y) indicator as a comparison between control (T1) and Demonstration (T2) treatments. Values are the mean of 12 replicate parcels and bars with the same letter are not significantly different at the 0.05 level of probability according to the Duncan test.

Similar trends as those mentioned for yield were also observed for the Water Productivity indicator (Table 3), with the highest differences being recorded during the dry year of 2018 in the rainfed orchards, where WP was 2.02 times higher in T2, as compared to T1. This case was the only one that showed statistically significant differences among treatments. However, WP was higher in the demonstration part of the orchards for all years and irrigation strategies, ranging from 1.08 to 2.02 times higher in T2 as compared to T1 (Figure 6). WP was also at its highest level of difference between T2 and T1 for irrigated orchards during 2018 (1.49 times higher). It has to be mentioned that irrigated orchards were mostly located in the Western pilot area, while the rainfed orchards were mostly

located in the Eastern pilot area, which was affected more severely during the dry year of 2018, having one of the lowest yields during the last 20 years (unpublished data from the local Directorate of Agricultural Economy and Veterinary Medicine). Table 3 indicates that in the studied parcels, the range of WP values was recorded to be between 2.02 Kg/m$^3$ and 8.53 Kg/m$^3$. These findings are in agreement with previous studies regarding the evaluation of water use efficiency for olive grove systems both in Greece [37,38] and other Mediterranean countries such as Tunisia, Jordan, Spain, Portugal, and Italy [7,39–43].

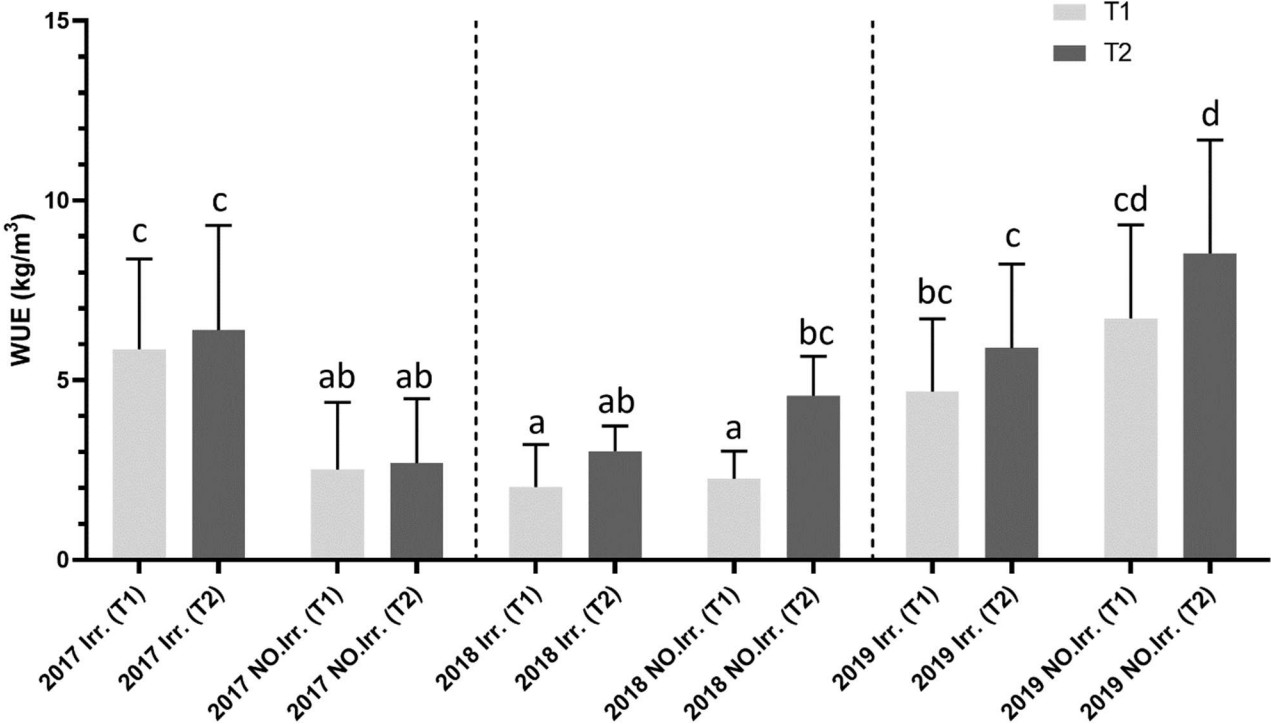

**Figure 6.** Water Productivity (WP) indicator as a comparison between Control (T1) and Demonstration (T2) treatment. Values are the mean of 18 replicate parcels and bars with the same letter are not significantly different at the 0.05 level of probability according to the Duncan test.

Despite the non-statistical differences in Y and WP, when more parameters were introduced in the calculation of the performance indicators, statistical differences were recorded. Hence, the introduction of the oil percentage of the fruit and actual product price in the calculation of the EWP indicator resulted in statistically significant differences in 2018 and 2019 (Table 3). During these years, EWP was higher in T2 by 2.00 to 2.20 times in irrigated and by 1.36 to 1.78 times in rainfed orchards, as compared to T1 (Figure 7). The highest EWP values were recorded for rainfed T2 orchards in 2019 as well as 2018, which is reasonable since 2019 was a wet year contributing to rainfed farms having high amounts of olive oil percentage in their fruits. Also, the pest and disease problems in 2019 significantly reduced olive oil production increases the price of the product. Regarding 2018, the case of rainfed T2 had the highest yield of all the rainfed cases, during a year with the lowest water input in the orchard due to the long dry season. Nevertheless, the use of production functions to assess olive EWP provides a broader view of farm profit than just the WP indicator, the results of EWP among different studies and different locations are hardly comparable. This is because EWP considers economic, social, and olive oil percentage values, which vary substantially from one season to another as well as from one location to another [27].

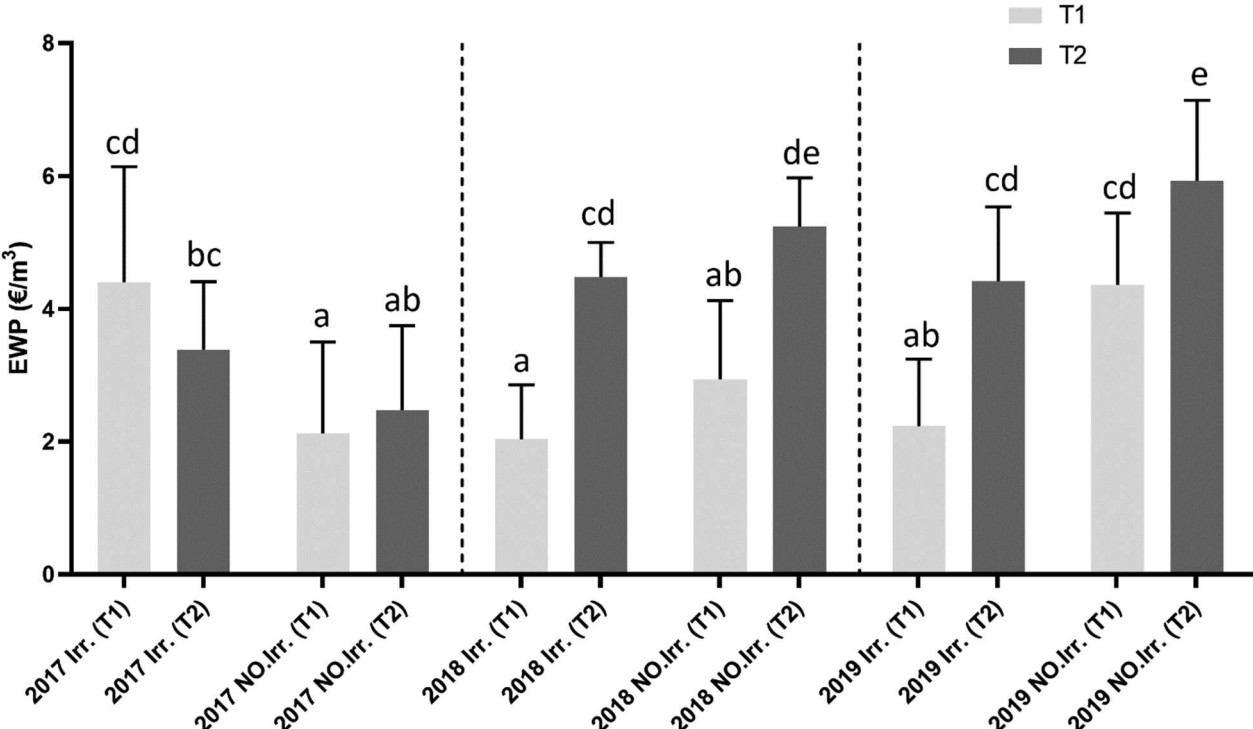

**Figure 7.** Economic Water Productivity (EWP) indicator as a comparison between Control (T1) and Demonstration (T2) treatment. Values are the mean of 18 replicate parcels and bars with the same letter are not significantly different at the 0.05 level of probability according to the Duncan test.

Concerning the runoff data from the four orchards in sloping areas, it was evident in most cases that the introduced barriers were effective in reducing the amount of runoff water (Table 3). Runoff water losses in T1 were up to 190% higher as compared to the amount of water lost in T2. The amount of runoff water collected in the traps was highest during the wet year of 2019 and lowest during the dry year of 2018 (Figure 8). As a percentage, the water saving (more water stored in the soil) was higher during the driest year (2018), which is reasonable since during the wet year (2019) the storage capacity of the soil was limited after continuous rainfall events. The above findings sound reasonable considering that Espejo-Pérez [44] found that the runoff reduction in rainfed sloping olive groves was up to 56% higher in the case of using cover crops (as the only measure for water retention) compared to conventional tillage practice. Also, experiments on steep soil olive groves in Italy clearly showed that the soil cover by mulching cover (residues from pruning) increased soil infiltration rates by over 100% compared to mechanical tillage traditional soil management practices [45].

Taking into consideration that 2018 was one of the driest years of the last two decades in Crete as well as that 2019 was one of the wettest years of the last twenty years is of high importance to investigate the efficiency of the applied practices to control water surface runoff in sloping land. Based on the results, during the three implementation years both and for all the studied sloping farms in the Control (T1) treatments much more water was collected compared to the Demonstration (T2) treatments (Surface Runoff reduction in demonstration parcels). An indication is that due to the natural barriers, that were placed at strategically selected locations (vertical to the flow) in sloping demonstration treatments, the water losses were reduced.

As mentioned in the methodology section, in order to quantify the effectiveness of the applied natural barriers in sloppy parcels, water traps were constructed to measure the amount of water collected both for Control (T1) and Demonstration (T2) treatments. The optimal plan is for the traps in the T2 treatments to collect lower amounts of water compared to the T1 treatments and this was verified by the results. This is also very important if we

think of the case of flash flood events, commonly occurring in semiarid Mediterranean environments [46]. The results of the present study concerning water retention could verify the hypothesis of large-scale studies [47], that if natural barriers are established at sloping farms of semi-mountainous areas (river basin scale) the amounts of runoff water that will reach downstream areas could be lower, mitigating the effects of floods in these areas.

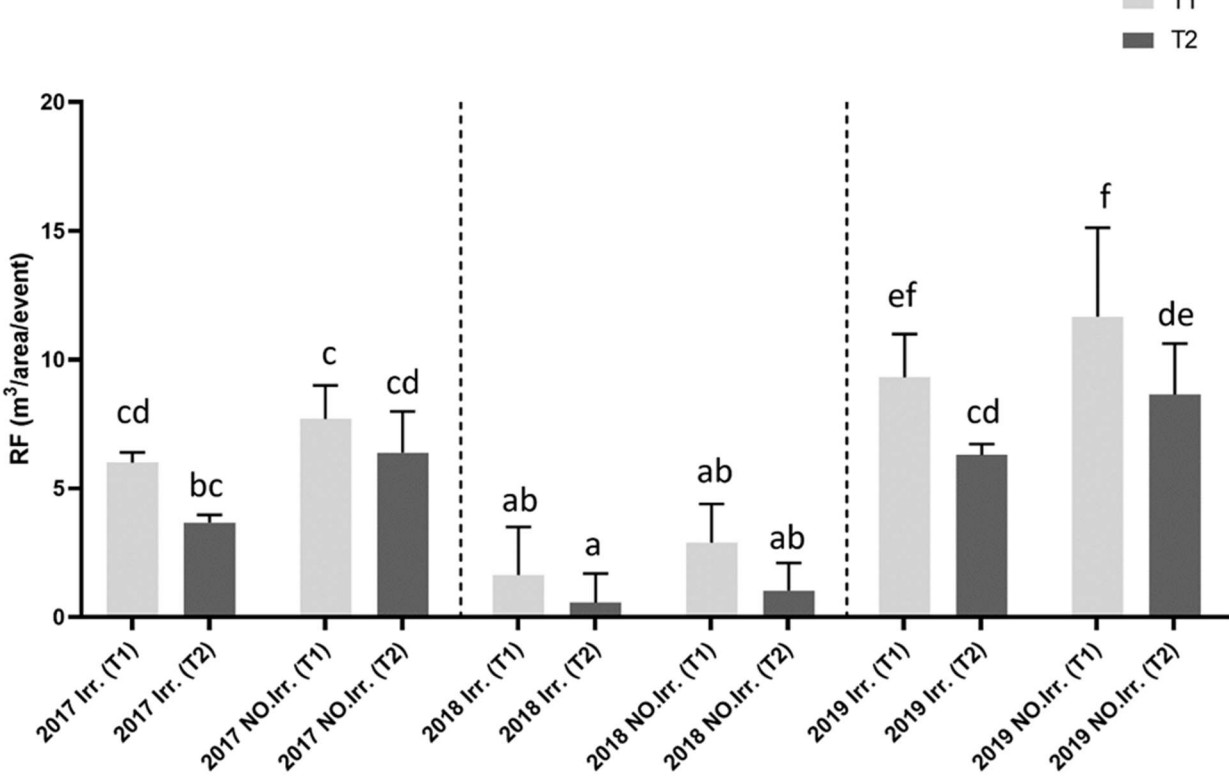

**Figure 8.** Runoff (RF) indicator (expressed in m³ per 0.2 ha per rainfall event) as a comparison between Control (T1) and Demonstration (T2) treatment. Values are the mean of four replicate parcels and bars with the same letter are not significantly different at the 0.05 level of probability according to the Duncan test.

An interesting finding, according to the results, is related to irrigation and no irrigation strategy. As shown in Figure 7, in rainfed sloping parcels the amount of water collected was greater than that of the irrigated parcels, regardless of the applied treatment. This can be explained due to the soil crust generated after dry periods (no irrigation applications) followed by autumn high rainfall intensities. Mainly in arid and semi-arid regions with poor soils (such as the case of semi-mountainous areas of Crete), surface runoff is frequently created as a result of the crust development on the soil surface [48]. The soil crust is a thin layer of higher shear strength and lowers hydraulic conductivity than the underlying soil. In dry summer conditions, soil crust can be generated at a high degree when a soil surface dries out after the wet period (winter and spring seasons). Thus, after dry periods, high rainfall intensities occur that fall over crusted parcel soils encouraging Horton-type overland flow causing a very fast response. This crust can affect, positively, the generation of surface runoff [49]. Since each of the proposed practices has a positive impact on farm water conservation, our aim was to suggest an integrated management plan for GAPs where its implementation could positively affect the sustainability of olive cultivation considering the effects of climate change. Furthermore, all the suggested practices can be easily adopted by farmers either in the case of non-sloping or sloping parcels.



## 4. Conclusions

In the present study, the results of four well-established indicators were analyzed in order to assess the efficiency of 13 Good Agricultural Practices (GAPs) related to water and soil, in increasing the adaptation performance of Mediterranean olive growing in extreme climate-water conditions. These GAPs were implemented in 18 experimental olive groves (irrigated and rainfed) which are located in Crete (south Greece), and they were divided into two treatments, namely the Control (T1, business-as-usual) and the Experimental (T2, GAPs implementation).

The water-related performance indicators, including Water Productivity (WP), Economic Water Productivity (EWP), and Runoff (RF), gave better overall results in the Demonstration (T2) than the Control (T1) Treatments. Since the beginning of the implementation of these practices in almost all cases the values of all studied indicators in T2 orchards were higher than in T1, with the biggest differences noted during the years of extreme conditions (2019 extremely wet, 2018 extremely dry). Through these indicators, the implementation of our practices shows positive results under different or even extreme climate conditions. Concerning the actual olive Yield achieved, during 2017 a slight increase was observed as this was the first year of implementation and such practices need time to show positive results. In the extremely dry year of 2018, there was a significant increase in yield in the demonstration parts (T2) compared to the control parts (T1), while under the opposite climate conditions of 2019 which was an extremely wet year and at the same time suffered intense pest infestation, we still managed to achieve an increase in yield.

The results of the present study indicate that the implementation of the GAPs improved water use efficiency. Considering that increasing water scarcity, droughts frequency, and climate extremes constitute the main threats to Mediterranean agriculture, our GAPs demonstrate significant potential to support the adaptability of the olive crop to extreme climate and water conditions. Thus, maintaining olive productivity in extreme climate conditions can be achieved through the implementation of GAPs involving better management of pruning, irrigation, and fertilization combined with long-term soil improvement planning and reduction of water and nutrient losses.

**Author Contributions:** Conceptualization, N.N.K.; Methodology, N.N.K., G.P. and G.A.; Software (data analysis and visualization), N.N.K.; Data curation, G.M. (Giasemi Morianou), G.K., N.D., S.M., K.A. and G.M. (Georgios Motakis); Writing-original draft preparation, N.N.K., G.P., G.M. (Giasemi Morianou), V.P. and G.A.; Writing-review and editing, N.N.K., G.K. and G.P.; Supervision, N.N.K. All authors have read and agreed to the published version of the manuscript.

**Funding:** This research received no external funding.

**Acknowledgments:** With the contribution of the LIFE+ financial instrument of the European Union, for the project LIFE14 CCA/GR/00389-AgroClimaWater.

**Conflicts of Interest:** The authors declare no conflict of interest.

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
