# Peer review of "Good Agricultural Practices Related to Water and Soil as a Means of Adaptation of Mediterranean Olive Growing to Extreme Climate-Water Conditions"

_sustainability, doi:10.3390/su142013673_

Round 1
Reviewer 1 Report
Authors have encountered 13 GAPs for higher productivity. It is very much difficult to separate out the most influential practice. Authors has been suggested to narrow down the GAPs, so that olive growers can easily adapted the improved production system.
Results were quite abrupt due to tremendous variations in the precipitation among the years. It is suggested to do pooled analysis of three consecutive years for better understanding and clarity.
Authors must include a brief description of soil properties.
Author Response
Comments
Dear Editor/ Reviewers,
The authors tried to consider all your valuable comments which have improved significantly the manuscript.
Responses are summarized in this document and are also highlighted in the manuscript in blue font.
Reviewer 1
Comment 1.
Authors have encountered 13 GAPs for higher productivity. It is very much difficult to separate out the most influential practice. Authors has been suggested to narrow down the GAPs, so that olive growers can easily adapted the improved production system.
Reply: Thank you for your comment. Each of the proposed practices has their positive impact on farm water conservation. However, our aim is not to separate which of the applied practices has the greatest effect, but to propose an integrated management plan of simple agricultural practices where its implementation could positively affect the sustainability of olive cultivation considering the effects of climate change. In addition, all the proposed practices can be easily adopted by the farmers either in the case of non-sloping or sloping parcels. The above definition was added in the discussion section (see lines 437-442)
Comment 2.
Results were quite abrupt due to tremendous variations in the precipitation among the years. It is suggested to do pooled analysis of three consecutive years for better understanding and clarity.
Reply: Thanks a lot for this comment. The analysis in the 3 different years was done precisely to show the effectiveness of the practices under different climatic conditions of the three consecutive years (average climate conditions for 2017, extreme dry climate conditions for 2018, and extreme wet climate conditions for 2019). If the analysis were done for all the years together (pooled analysis of three consecutive years), no conclusions could be drawn about the extreme climatic conditions, which was our goal. The above definition was added in the methodology section (see lines 262-272)
Comment 3.
Authors must include a brief description of soil properties
Reply: Excellent comment. The suggestion was taken into consideration, a table of the main soil properties of the experimental plots was added to the text (see Table 1)
Reviewer 2 Report
In annex.

Author Response
Comments
Dear Editor/ Reviewers,
The authors tried to consider all your valuable comments which have improved significantly the manuscript.
Responses are summarized in this document and are also highlighted in the manuscript in blue font.
Reviewer 2
Comment 1.
Line 171. Seven out of 18 studied farms were irrigated and the remaining 11 were rainfed.
Reply: The above suggestions were taken into consideration
Line 231. TWU = irrigation + rainfall, Comment: Instead of the word rainfed, please use rainfall or precipitation
Reply: TWU is the amount of water that were obtained from both rainfall events and irrigation amounts during the irrigation period (May to October). Please consider that, in the text where you came across the word “rainfed” it is to describe the non-irrigated parcels, i.e. those that are watered only by rain.
Rainfed agriculture is a common scientific term that describes a type of farming that relies on rainfall for water (for example, please see the title of the following article: Long-term changes in rainfed olive production, rainfall and farmer’s income in Bailén (Jaén, Spain). Euro-Mediterr. J. Environ. Integr. 6, 58. https://doi.org/10.1007/s41207-021-00268-1)
Line 181. Figure 2 should be removed because it does not present any relevant information. The description of the experiment is presented in the text of the article.
Reply: The authors believe that Figure 2 depicts a graphical representation of the experimental design that helps the reader to understand immediately its application in the field.
Line 182-210. The lines should be deleted because similar information is given in table 1.
Reply: Thank you for your comment. The lines deleted according to your suggestion.
Line 222. Figure 3 should be removed because it does not present any relevant information.
Reply: We believe that this particular Figure gives the reader the ability to understand how the different components of the hydrological cycle interact at scale of parcel/farm. Most readers have associated the concept of the hydrologic cycle only at the watershed level and they are not familiar with the effects of the different water cycle components on farm scale, so this figure could give a comprehensive representation of this prospect.
Line 223. Please indicate what irrigation system was used (was it artificial rain with sprinklers). Please provide the amount of water supplied to the plantations (m3 *ha-1 ). This will make it easier for the reader to calculate the TWU.
Reply: Thank you for your excellent comment, we added information about irrigation is lines 191-201.
Line 299. In Table 2, provide parameter values for non-sloping and sloping parcels. Otherwise, delete the table as the same results are shown in Figures 5-8.
Reply: The authors believe that this table (in the revised version Table 3) gives valuable information as it summarizes the results of the studied Yield and Water use performance indicators for all the years. In addition, it presents a numerical information not seen in the corresponding figures and helps to the discussion of the results.
Lines 280-290. The climatic conditions are presented very generally. A detailed description is required in the additional chapter: Climate Conditions and the Impact on Cultivation Technology
Reply: The suggestion was taken into consideration and new information was given in the text about climate and soil conditions (please see lines: 120 – 124, 262-267 and lines 152 - 160)
Line 31. Rainfall not rainfed.
Reply: Please consider as mentioned above that the word “rainfed” refers to the non-irrigated plots.
Line 316. This is supported by the fact that 2018 and 2019 were two years with high and low extremes on precipitation. The sentence is incomprehensible without the given climatic conditions.
Reply: Please this information is reported on lines 262-267
Figure 5-8. Please make box plots in the Statistical software
Reply: Thank you for your comment. In general, bar graphs are suitable for counts/values, while box plots are used to represent the characteristics of a distribution. In this work, our goal was to plot our values (mean indicator values) in a simple and efficient way and not to give characteristics of a distribution. In addition, we chose the bar graph type to represent our results, as it depicts, among other things, the important information of the standard deviation. Finally, bar graphs are easier to understand by non-statistical readers. Thank you for your understanding.
Round 2
Reviewer 2 Report
The reviewer's comments were taken into account.